# A Predictive Model of the Start of Annual Influenza Epidemics

**DOI:** 10.3390/microorganisms12071257

**Published:** 2024-06-21

**Authors:** Elisabet Castro Blanco, Maria Rosa Dalmau Llorca, Carina Aguilar Martín, Noèlia Carrasco-Querol, Alessandra Queiroga Gonçalves, Zojaina Hernández Rojas, Ermengol Coma, José Fernández-Sáez

**Affiliations:** 1Primary Care Intervention Evaluation Research Group (GAVINA Research Group), IDIAPJGol Terres de l’Ebre, 43500 Tortosa, Spain; ecastro@idiapjgol.info (E.C.B.); ncarrasco@idiapjgol.info (N.C.-Q.); aqueiroga@idiapjgol.info (A.Q.G.); zhernandezr.ebre.ics@gencat.cat (Z.H.R.); jfernandez@idiapjgol.info (J.F.-S.); 2Campus Terres de l’Ebre, Universitat Rovira i Virgili, 43500 Tortosa, Spain; 3Terres de l’Ebre Research Support Unit, Foundation University Institute for Primary Health Care Research Jordi Gol i Gurina (IDIAPJGol), 43500 Tortosa, Spain; 4Servei d’Atenció Primària Terres de l’Ebre, Institut Català de la Salut, 43500 Tortosa, Spain; 5Unitat d’Avaluació, Direcció d’Atenció Primària Terres de l’Ebre, Institut Català de la Salut, 43500 Tortosa, Spain; 6Primary Healthcare Information Systems, Health Institute of Catalonia, 08007 Catalonia, Spain; ecomaredon@gencat.cat; 7Unitat de Recerca, Gerència Territorial Terres de l’Ebre, Institut Català de la Salut, 43500 Tortosa, Spain; 8Unitat Docent de Medicina de Familia i Comunitària, Tortosa-Terres de l’Ebre, Institut Català de la Salut, 43500 Tortosa, Spain

**Keywords:** influenza, syndromic surveillance, public health, predictive modeling

## Abstract

Influenza is a respiratory disease that causes annual epidemics during cold seasons. These epidemics increase pressure on healthcare systems, sometimes provoking their collapse. For this reason, a tool is needed to predict when an influenza epidemic will occur so that the healthcare system has time to prepare for it. This study therefore aims to develop a statistical model capable of predicting the onset of influenza epidemics in Catalonia, Spain. Influenza seasons from 2011 to 2017 were used for model training, and those from 2017 to 2018 were used for validation. Logistic regression, Support Vector Machine, and Random Forest models were used to predict the onset of the influenza epidemic. The logistic regression model was able to predict the start of influenza epidemics at least one week in advance, based on clinical diagnosis rates of various respiratory diseases and meteorological variables. This model achieved the best punctual estimates for two of three performance metrics. The most important variables in the model were the principal components of bronchiolitis rates and mean temperature. The onset of influenza epidemics can be predicted from clinical diagnosis rates of various respiratory diseases and meteorological variables. Future research should determine whether predictive models play a key role in preventing influenza.

## 1. Introduction

Influenza causes epidemics in the cold season of the year, during which between 290,000 and 650,000 people die throughout the world each year [1]. Worldwide and at the European level, the World Health Organization recommends monitoring this disease using sentinel networks in all countries. Sentinel networks exist to identify circulating respiratory viruses and estimate their incidence [2].

Spain has a surveillance system that collects data provided by the sentinel network of each of its 19 autonomous communities and cities [3]. All the information obtained by the sentinel network of Catalonia is published on an open access website, Sistema d’Informació per a la Vigilància d’Infeccions a Cataluña (SIVIC) [4]. This website also contains the frequencies of clinical diagnoses of a range of respiratory diseases registered in computerized clinical histories of primary care.

Annual epidemics are associated with the presence of other respiratory viruses, such as a syncytial respiratory virus (which usually precedes influenza) [5], pneumovirus, and parainfluenza [6]. The relationship between influenza and meteorological factors has also been studied. Generally, low temperatures and low absolute humidity are associated with a higher incidence of influenza [7,8,9], although this pattern is not as evident in the tropics because of the narrow range of temperature variation in the region.

These variables have been used in various kinds of models to predict influenza epidemics. In this study, we compared two groups: statistical and automated learning models. Some of the statistical models used for influenza prediction are ARIMA (autoregressive integrated moving average) models [10] and generalized linear models (GLMs). The family of GLMs includes quasi-Poisson [7], negative-binomial [11], and functional-regression models, among others [12]. This type of model focuses on predicting the influenza rate several weeks ahead as accurately as possible. However, these models do not predict the time of onset, when the rate begins to increase exponentially. Random Forest [13], Support Vector Machine [14], and Deep Learning [15] models are the automated-learning models used most often for predicting influenza epidemics.

Influenza epidemics place a strain on healthcare systems, increasing the volume of visits. A statistical model capable of predicting influenza epidemics would help optimize healthcare, resource management, and preventive strategies. Therefore, the main aim of this study was to construct a model capable of predicting influenza epidemics at least one week in advance, using clinical diagnostic rates of respiratory diseases and meteorological variables.

## 2. Materials and Methods

### 2.1. Design and Study Population

We conducted a population-based ecological time-series study, using rates of clinical diagnosis of different respiratory diseases and meteorological variables. The study period ran from week 40 of 2011 to week 20 of 2019 (8 seasons). The influenza season was defined as the period between week 40 of a particular year and week 20 of the following year, although interseason data were also used.

### 2.2. Data Collection

Clinical diagnoses in Primary Care are based on suspicion and most are not virologically confirmed. The respiratory diseases considered were bronchiolitis, influenza, other acute respiratory infections (ARIs), and all-causes pneumonia. The diagnostic codes included in the study for each respiratory disease were (for more detail, see Appendix A):-Bronchiolitis: J21.0, J21.8, J21.9.-Influenza: J9–J11.-Other ARIs: J00, J04, J02.9, J03.9, J06.9, J20.3–J20.9.-All-causes pneumonia: J12, J17.1, J18.8, J18.9.

The number of clinical diagnoses of different respiratory diseases was obtained from the SIVIC website [4]; these are publicly available secondary data. SIVIC integrates the information collected in primary care centers, hospitals, laboratories, and the Public Health Agency of Catalonia, allowing the analysis of acute respiratory infections in real time to monitor trends and provide alerts. We calculated weekly diagnostic rates for each respiratory disease in Catalonia.

Virologically confirmed cases of influenza were not included as a previous study evidenced that clinical diagnosis rates of influenza are equivalent to virologically confirmed rates. These studies analyzed the concordance between both influenza surveillance systems and evaluated which of these systems could provide the earliest detection of the start of the influenza epidemic [16,17].

We calculated the epidemic threshold for Catalonia from weekly influenza rates for each season using the Moving Epidemic Method (MEM) [18]. The MEM method determines the baseline for influenza activity and establishes an epidemic threshold. This epidemic threshold was used to create the dependent variable by comparing the weekly influenza diagnosis rate with the calculated epidemic threshold. We consider an influenza epidemic to have arisen when the rate is higher than the threshold.

We collected data from 163 MeteoCat automatic weather stations [19], which are published in Portal de Dades Obertes de Catalunya [20]. For each station, we downloaded diary data of the mean, minimum and maximum temperatures and the relative humidity. Absolute humidity was calculated using mean temperature and relative humidity.

To obtain a weekly average of the meteorological variables, we weighted the weather data by the population under the influence of each station. To this end, we assigned each healthcare center to the corresponding municipality based on the list of centers with recorded activity during the study period. The method for assigning a reference weather station involved selecting the station based on parameters such as proximity, altitude, and similar geographic and climatic characteristics. We calculated the population under the influence of each weather station by grouping the population of the municipalities (population data taken from the central population registry of CatSalut, downloaded from the Portal de Dades Obertes de Catalunya [20]) with healthcare centers assigned to that station. To calculate the weekly averages of meteorological data for Catalonia, we weighted them by the population under the influence of each station. The station assignation to each healthcare center was validated by experts.

We calculated the respiratory infection rate per 100 000 inhabitants in Catalonia. The annual population of Catalonia was retrieved from IDESCAT (Institut d’Estadística de Catalunya) [21].

### 2.3. Statistical Analysis

The selected models were of the logistic regression, Support Vector Machine, and Random Forest types. For automated learning methods, hyperparametric tuning was carried out (Appendix B).

As independent variables, we included the diagnostic rates of various respiratory diseases (excluding influenza), and meteorological variables from previous weeks (lagged variables) up to the week of the dependent variable. We used maximum likelihood estimation to decide the variables for inclusion in each model.

A principal component analysis (Phyton sklearn.descomposition.PCA [22]) of the clinical diagnoses and meteorological lagged variables was conducted. Two principal components (PCs) were obtained from each diagnosis and the meteorological variables. We used the first six seasons for training and the final two for internal validation of the models. Finally, the performance of each model for the validation dataset was evaluated using the Kappa index, the Area Under the ROC Curve (AUC), and the accuracy between the values predicted by the models and the actual values of the validation dataset. A predictive index was estimated from the logistic regression.

Statistical analyses were performed using R version 4.2.2 and Python version 3.11.4.

## 3. Results

Our study shows that the PC logistic regression model was the most accurate and had the highest Kappa index. The Support Vector Machine model had the highest AUC value. The Kappa index was high in all models, and the PC logistic regression and Support Vector Machine approaches both yielded narrower confidence intervals clustered around a value of 1 (Table 1).

The median predictive index in the logistic regression without PCs was closer to 100 than the PC logistic regression. However, the interquartile range was wider than in the case of the logistic regression involving the PCs (Box 1). The AUC was greater than 0.950 in all models, and no statistically significant differences were found between the models.

Box 1Predictive index of logistic regression models.Logistic Regression without PCs      
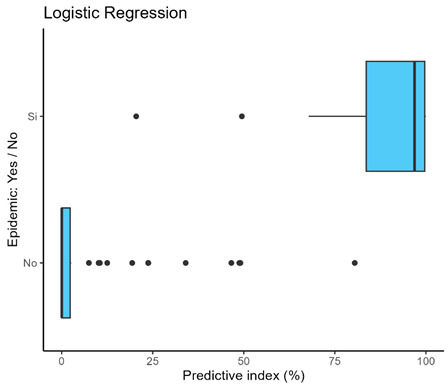


$$Predictive\;Index=\frac{e^{0.4595\;X_{1}+0.1522X_{2}+0.0032X_{3}-0.5441X_{4}-0.2895X_{5}-1.8188}}{e^{0.4595\;X_{1}+0.1522X_{2}+0.0032X_{3}-0.5441X_{4}-0.2895X_{5}-1.8188}+1}\times 100Predictive\;Index=\frac{e^{0.7126\;X_{1}-0.3457X_{2}+0.05363X_{3}-1.7687X_{4}}}{e^{0.7126\;X_{1}-0.3457X_{2}+0.05363X_{3}-1.7687X_{4}}+1}\times 100$$

Logistic Regression with PCs      
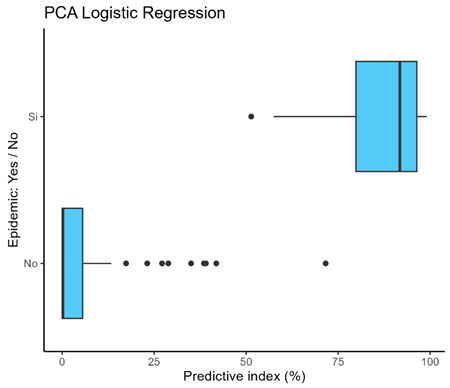


$$Predictive\;Index=\frac{e^{0.7126\;X_{1}-0.3457X_{2}+0.05363X_{3}-1.7687X_{4}}}{e^{0.7126\;X_{1}-0.3457X_{2}+0.05363X_{3}-1.7687X_{4}}+1}\times 100$$



In both logistic regression models, an increase in the bronchiolitis rate was associated with a significant rise in the risk of influenza epidemics, while increased mean temperature appeared to protect against epidemics (Table 2 and Table 3). The detailed table for the logistic regression model can be found in Appendix C.

Selected variables for the Support Vector Machine model were Bronchiolitis PC1, Bronchiolitis PC2, Pneumonia PC1, Mean Temperature PC1, and Absolute Humidity PC2.

Variables selected for the Random Forest model included Other ARI CPs that did not feature in other models. The most important variables in this model were Mean Temperature PC1, Other ARI PC2, and Bronchiolitis PC1 (Figure 1).

## 4. Discussion

This study constructed a PC logistic regression model capable of predicting, at least one week in advance, the onset of influenza epidemics using clinical diagnoses of respiratory diseases and meteorological variables from previous weeks. This model performed best of those evaluated in terms of its accuracy and predictive index. The bronchiolitis and mean temperature PCs were the most significant variables in the model. All the models’ results were excellent, and no statistically significant differences were found between the performance metrics across the models. However, the PC logistic regression model yielded higher Kappa and punctual accuracy estimates than did the other models. For this reason, we chose this model to predict the onset of influenza epidemics as accurately as possible.

The choice of optimal model to predict an event varies between studies. Such selection depends on the outcome variable, its components, the early acquisition of data, and data quality. The outcome variable was calculated by the MEM method. It is expressed as a dichotomous variable for the week of onset of an influenza epidemic, thereby defining the start of the epidemic in a study season. In other words, the variable provides information about whether the influenza rate for that week will exceed the threshold defining epidemic onset. Similarly, some studies examined the probability of epidemics in future weeks using Markov models. These models are less explanatory but provide probabilities well in advance of epidemics [23]. Another approach with a similar goal has involved calculating the point at which the trend changes and the slope begins to increase, using methods such as Bayesian Online Change Point [24], joinpoint regression [17], and others [25]. These methods enable epidemics to be detected when the number of cases begins to rise, long before the epidemic threshold is reached, although it cannot predict if such a change will mark the onset of the influenza epidemic [17]. In contrast, the MEM method does not anticipate as far into the future but ensures that the epidemic’s onset is correctly determined.

Several studies aiming to predict influenza case rates several weeks ahead have used methods other than logistic regression because they have employed a continuous outcome variable [26,27]. However, for dichotomous outcome variables, logistic regressions have been used to predict the risk of a specific event occurring [28]. For example, a model predicting air quality based on pollution levels yielded very good results that were comparable to those emerging from neural networks and Support Vector Machines [29]. This emphasizes how the characteristics of the outcome variable are crucial when selecting optimal study models and for addressing the goal of our study. Logistic regression models with and without PCs are considered classic prediction models and are proven to have good predictive capacity. Two new models were developed to identify potential areas for improvement.

Classic predictive models are often compared with machine learning models. Random Forest and Support Vector Machine models have been used to predict influenza rates, yielding good results and providing better error rates than those obtained from classic methods such as ARIMA [30,31]. In comparative studies, predictions from Support Vector Machines were more accurate than those produced by Random Forest models [31]. However, the confidence intervals from the Random Forest approach were more robust. Our results confirm that the Support Vector Machine has a higher accuracy rate than the Random Forest model, but we have not been able to compare the robustness of the confidence intervals.

In selecting the model, consideration was also given to the availability and immediacy of data acquisition since early prediction requires immediate data collection. Likewise, a data source, such as SIVIC, that is updated weekly based on clinical diagnoses is a key tool for ultimately implementing a prediction equation. In this regard, diagnoses extracted from medical records regarding the number of influenza cases have been used on several occasions to predict the number of influenza cases in subsequent weeks [32].

Conversely, daily values of temperature variables are available in the MeteoCat database, making it optimal for predicting the onset week of the annual influenza epidemic sufficiently in advance. The temperature during the weeks of autumn has been identified in various studies as a factor associated with the onset of the influenza epidemic. Our results indicate that a drop in temperature raises the risk of an epidemic. This is consistent with the findings of other studies conducted in South Korea [8], Canada [11], the Netherlands [33], Japan [34], and China [9] that demonstrate increases in influenza incidence as temperature and absolute humidity decrease.

The other main component of the equation was the bronchiolitis rate. An increase in bronchiolitis incidence can predict an increase in influenza cases in the following weeks [5]. This association could be explained by the interaction between respiratory viruses. Several studies have shown that Influenza A and the respiratory syncytial virus interact negatively, meaning that the decline in the respiratory syncytial virus peak could predict the rise of the influenza peak [35,36].

Additionally, both logistic regression models include the all-causes pneumonia rate and, in the case of logistic regression without PCs, the rate of other acute respiratory infections. These two variables may be clinically significant and be of value to healthcare professionals. Both logistic regression models include variables that are not significant in themselves but whose exclusion significantly reduces model accuracy.

The main strength of this study is the high accuracy obtained in the internal validation of each of the models, allowing the prediction of the influenza epidemic onset with one week’s notice. Furthermore, the variables on which the models are based are easy to understand and are published on open-access websites. Implementing an influenza epidemic prediction model could be crucial for healthcare system preparedness and epidemic management. The utility of the model could be reinforced by combining it with another that provides information about the influenza rate for the following week. This would allow the intensity and peak week of the epidemic to be predicted.

There are several limitations of this study. First, the pre-pandemic seasonal models may differ from post-COVID-19 pandemic models. Second, influenza diagnoses are based on suspicion and are not confirmed by laboratory tests. The SIVIC database compiles data from syndromic surveillance (clinical diagnosis of influenza) and sentinel surveillance (virological confirmed cases of influenza) but is unable to discern how many suspected cases have been confirmed by testing [16]. Moreover, it is worth noting that suspected influenza diagnosis rates in primary care coincide with confirmed rates [16], and there is no time lag between them [17]. Respiratory infection data are obtained with a one-week delay. Obtaining them daily from records of the previous day would be very helpful for making our predictions, as with meteorological data.

A third limitation is the use of aggregated meteorological data for a large geographical area featuring a range of climates. To mitigate this problem, the population under the influence of each station was considered so that the meteorological data considered in the model represent a large part of the population with respect to altitude, proximity, and absence of geographical barriers with the reference station.

In future studies, models obtained will need to be validated against post-COVID-19 respiratory infection and meteorological data and in other regions of Europe. It is also necessary to evaluate the model performance in real-time, and subsequently to run a pilot test on an open platform, like SIVIC, before finally implementing it fully. Furthermore, regarding meteorology could be interesting to apply to smaller regions in order to know if local variations in weather can impact influenza transmission or other respiratory diseases.

## 5. Conclusions

A model has been derived that allows the onset of an influenza epidemic to be predicted with at least one week’s notice using logistic regression with principal components. The accuracy, Kappa, and AUC values obtained from internal validation are high. The main principal components were bronchiolitis behavior and temperature in the previous weeks. Future studies will need to validate model performance in other regions and in post-pandemic seasons and to investigate whether predicting the onset of influenza epidemics could have implications for resource management of healthcare systems.

## Figures and Tables

**Figure 1 microorganisms-12-01257-f001:**
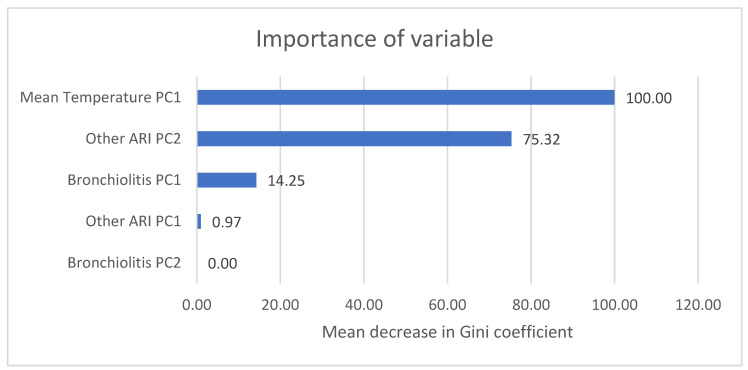
Variable importance for Random Forest model.

**Table 1 microorganisms-12-01257-t001:** Model results.

Model	Kappa (95% CI)	Area Under Curve (AUC)	Accuracy	Parameters
Logistic regression without PCs	0.897 (0.784, 1.000)	0.990 (0.974, 1.000)	0.955	
Logistic regression with PCs	0.933 (0.842, 1.000)	0.996 (0.988, 1.000)	0.986	
Support Vector Machine	0.901 (0.793, 1.000)	0.998 (0.994, 1.00)	0.959	C = 5
Random Forest	0.793 (0.636, 0.951)	0.983 (0.961, 1.000)	0.918	n trees = 200 mtry= 2

PCs: principal components. C: cost parameter of Support Vector Machine. Mtry: number of variables in each division.

**Table 2 microorganisms-12-01257-t002:** Logistic regression model without PCs.

Variable	Lower Risk of Epidemics	1	Higher Risk of Epidemics	OR	95% CI		*p*
Bronchiolitis rate 6 weeks before	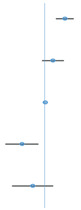			1.58	1.29	1.95	<0.001
Pneumonia rate 2 weeks before				1.20	0.93	1.56	0.169
Other ARI rate 4 weeks before				1.00	1.00	1.01	0.409
Mean temperature 3 weeks before				0.58	0.39	0.86	0.006
Absolute humidity 4 weeks before				0.75	0.46	1.21	0.241

ARI: Acute respiratory infection.

**Table 3 microorganisms-12-01257-t003:** Logistic regression model with PCs.

Variable	Lower Risk of Epidemics	1	Higher Risk of Epidemics	OR	95% CI		*p*
Bronchiolitis Principal Component 1	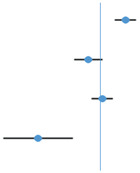			2.05	1.51	2.77	<0.001
Bronchiolitis Principal Component 2				0.71	0.47	1.06	0.097
Pneumonia Principal Component 1				1.06	0.79	1.42	0.723
Mean Temperature Principal Component 1				0.17	0.06	0.46	<0.001

Principal component: Linear combination of aggregator variables. The first component explains an important part of the aggregator’s variability (e.g., VRS).

## Data Availability

The original data presented in the study are openly available in SIVIC at https://sivic.salut.gencat.cat/ (accessed on 3 September 2021) and in Portal Dades Obertes de Catalunya at https://analisi.transparenciacatalunya.cat/ (accessed on 3 July 2020).

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
