# Peer review of "A Predictive Model of the Start of Annual Influenza Epidemics"

_microorganisms, 2024, doi:10.3390/microorganisms12071257_

Round 1

Reviewer 1 Report

Comments and Suggestions for Authors

Dear authors,

I have now completed the review of the manuscript titled "Predictive model of the start of annual influenza epidemics."

The research demonstrates a promising approach for influenza epidemic prediction.

The manuscript is interesting and, in general, fairly well-written. The paper aims to develop a model to predict the onset of influenza epidemics using data on rates of other respiratory diseases and meteorological variables. Being able to predict influenza epidemics in advance could be very valuable for healthcare system preparedness. 

Also, the study compares the performance of multiple predictive modeling approaches, including logistic regression with and without principal components, support vector machines, and random forests. This allows the authors to identify the best performing model.

The logistic regression model achieved high accuracy, Kappa, and AUC values in internal validation, suggesting it has good predictive ability. The key predictive variables identified, bronchiolitis rates and temperature, have clinical relevance.

Finally, the data used, including respiratory disease diagnosis rates and meteorological data, are publicly available which facilitates model implementation.

However, I still have some suggestions to further improve the quality of the manuscript.

I would like to suggest that the authors address these limitations in the article, either by discussing them in the limitations section or, where feasible, by making the appropriate revisions:

1. Aggregating low-resolution meteorological data over a large geographic area like Catalonia may obscure local variations in weather that impact influenza transmission. Authors may use higher geographic resolution to improve the models.

2. Since the influenza model incorporates meteorological variables, seeing how atmospheric conditions impact another respiratory condition could be informative. For example, recent population-based cohort studies on Atmospheric environment and persistence of pediatric asthma showed environmental factors associated with asthma persistence in children. In many countries, research on Effects of meteorological factors and air pollutants on the incidence of COVID-19 were done.

3. The models are based on pre-COVID-19 pandemic data. Respiratory disease dynamics may have shifted post-pandemic, so the models can be questioned.

4. Looking at patterns of other respiratory infections like COVID-19, may shed light on factors affecting transmission and spread of respiratory viruses in general, which could be relevant for influenza modeling. Comparing various research including incidence of healthcare utilization for allergic and respiratory infectious diseases in children with asthma before and during the COVID-19 pandemic may be useful.

Thank you for your valuable contributions to our field of research. I look forward to receiving the revised manuscript.

Author Response

First, thanks for your valuable commentaries. We are sure that the article will improve. Regarding to your commentaries,

  1. Aggregating low-resolution meteorological data over a large geographic area like Catalonia may obscure local variations in weather that impact influenza transmission. Authors may use higher geographic resolution to improve the models

We collected data from 163 meteorological stations across Catalonia and processed it to ensure it was representative of the region, considering the population in the influence area of each station. This approach allows the model to be applied and adjusted in the future using data from smaller geographic areas, such as basic health areas (aggregations of primary care centers).

We addressed this limitation in the Discussion Section (lines 277-281). Additionally, we incorporated your suggestion as future work (please see lines 285-288).

  1. Since the influenza model incorporates meteorological variables, seeing how atmospheric conditions impact another respiratory condition could be informative. For example, recent population-based cohort studies on Atmospheric environment and persistence of pediatric asthma showed environmental factors associated with asthma persistence in children. In many countries, research on Effects of meteorological factors and air pollutants on the incidence of COVID-19 were done

We added your suggestion as future work (please, see lines 286-288).

  1. The models are based on pre-COVID-19 pandemic data. Respiratory disease dynamics may have shifted post-pandemic, so the models can be questioned

This is completely true. We discussed it in the Discussion Section (lines 282-283), and we are currently readjusting our models using post-pandemic seasons. Additionally, this point is addressed in the future work section (see lines 282-283).

  1. Looking at patterns of other respiratory infections like COVID-19, may shed light on factors affecting transmission and spread of respiratory viruses in general, which could be relevant for influenza modeling. Comparing various research including incidence of healthcare utilization for allergic and respiratory infectious diseases in children with asthma before and during the COVID-19 pandemic may be useful

COVID-19 does not seem to have marked seasonality, unlike other respiratory viruses. However, studying the behavior of COVID-19 in terms of transmission and spread could be interesting, as well as examining the interactions between respiratory viruses.

Reviewer 2 Report

Comments and Suggestions for Authors

The author "present" a predictive model for influenza. The biggest weakness of the paper is that I was not able to see the model. This may be my own issue as I do not work in the very exact research subarea, but I was expecting to see a formula that will take several variables (such as temperature or bronchitis count) and predicts the influenza cases several weeks in advance. While this specific form of a model may not be possible, I would still welcome the authors to be specific and explain to the reader what the model is and how to use it

The second major weakness is that the model has not been validated at all. Can you run it on data not used to train the model? Even if it means data from the same region. However, the results would be much stronger if you took data from another region (or even another country) and the model was still working well

I also do not think the discussion is written appropriately. For example, the authors write "Similarly, some studies examined the probability of epidemics in future weeks using Markov models. These models are less explanatory but provide probabilities well in advance of epidemics". I would say that the sentences like that fit the intro better as they provide the background of the current research landscape. In the discussion, there should be some sort of a comparison of the old models versus the proposed model. While I understand that "These models are less explanatory" is some sort of a comparison, it would be much stronger if the comparison was quantitative (i.e. how those old models perform on the authors' data compared to the new and proposed models) 

Author Response

First, thanks for your valuable commentaries. We are sure that the article will improve. Regarding to your commentaries,

The biggest weakness of the paper is that I was not able to see the model. This may be my own issue as I do not work in the very exact research subarea, but I was expecting to see a formula that will take several variables (such as temperature or bronchitis count) and predicts the influenza cases several weeks in advance. While this specific form of a model may not be possible, I would still welcome the authors to be specific and explain to the reader what the model is and how to use it.

The probability of flu epidemics can be calculated from logistic regression models using the predictive index formula (Box 1). The predictive index formula is derived from the estimates of the model (Appendix 9C). Additionally, Tables 2 and 3 show the odds ratios for each variable, and Appendix C includes the model’s estimates, which can be calculated from the odds ratios using the following relationships: Estimate = ln(Odds Ratio) or Odds Ratio = e^Estimate.

Random forest models do not have a formula. Instead, these models consist of hundreds of different decision trees, each providing a prediction. The final prediction is the most voted option (Flu epidemics coming next week / No flu epidemics coming next week). Figure 1 shows which variables have the most predictive power. We used the R function randomForest from the randomForest package.

Support Vector Machines (SVM) work by finding the optimal boundary to separate different classes in the data, even in cases where the data is not linearly separable. However, SVMs do not provide the equation of this hyperplane. The variables selected to separate the classes are stated in the article (see lines 177-178). We used the R function tune from the e1071 package.

The second major weakness is that the model has not been validated at all. Can you run it on data not used to train the model? Even if it means data from the same region. However, the results would be much stronger if you took data from another region (or even another country) and the model was still working well.

We trained all the models using data from the 2011/2012 to 2016/2017 seasons and validated them with data from the 2017/2018 to 2018/2019 seasons, ensuring that the validation data were not used in training. We could not use data from other regions as it is not available online.

I also do not think the discussion is written appropriately. For example, the authors write "Similarly, some studies examined the probability of epidemics in future weeks using Markov models. These models are less explanatory but provide probabilities well in advance of epidemics". I would say that the sentences like that fit the intro better as they provide the background of the current research landscape. In the discussion, there should be some sort of a comparison of the old models versus the proposed model. While I understand that "These models are less explanatory" is some sort of a comparison, it would be much stronger if the comparison was quantitative (i.e. how those old models perform on the authors' data compared to the new and proposed models) 

Previous studies have focused on predicting flu rates several weeks in advance. However, the performance metrics of these models, such as Root Mean Squared Error (RMSE) or Mean Squared Error (MSE), are not comparable with the results of our study. In our study, the response variable is a categorical variable (flu epidemics coming next week / no flu epidemics coming next week). Therefore, the performance metrics used are Kappa Index, area under the ROC curve (AUC), and accuracy. These indicators range from 0 to 1, whereas RMSE and MSE range from 0 to infinity. This difference makes it impossible to compare the performance of our model with that of previously published models.

Reviewer 3 Report

Comments and Suggestions for Authors

In this manuscript, authors mainly constructed a PC logistic regression model for predicting the onset of annual influenza epidemics.Basically, it is of considerable interest, and this study is well designed and written, but there are some concerns for improving this manuscript.

1. Influenza seasons from 2011-2017 were used for model training, and those from 2017-2018 were used for validation. Why not use more another new data (2018-2024?)for validation?

2. As for predictive model of annual influenza epidemics, there were a lot of similar published studies, and authors should compare their work and provide the discrepancy with these papers.

3. The quality of English writing needs improvement. Specifically, there are so many very short paragraph.

Comments on the Quality of English Language

The quality of English writing needs improvement. Specifically, there are so many very short paragraph.

Author Response

First, thanks for your valuable commentaries. We are sure that the article will improve. Regarding to your commentaries,

  1. Influenza seasons from 2011-2017 were used for model training, and those from 2017-2018 were used for validation. Why not use more another new data (2018-2024?) for validation?

We stopped our study in 2019 due to the COVID-19 pandemic. Data from the COVID-19 pandemic period was very noisy and completely different. However, we are currently readjusting our models using data from 2019 to 2024. Your commentary is included as future work (see lines 282-283).

  1. As for predictive model of annual influenza epidemics, there were a lot of similar published studies, and authors should compare their work and provide the discrepancy with these papers.

Previous studies have focused on predicting flu rates several weeks in advance. However, the performance metrics of these models, such as Root Mean Squared Error (RMSE) or Mean Squared Error (MSE), are not comparable with the results of our study. In our study, the response variable is a categorical variable (flu epidemics coming next week / no flu epidemics coming next week). Therefore, the performance metrics used are Kappa Index, area under the ROC curve (AUC), and accuracy. These indicators range from 0 to 1, whereas RMSE and MSE range from 0 to infinity. This difference makes it impossible to compare the performance of our model with that of previously published models.

  1. The quality of English writing needs improvement. Specifically, there are so many very short paragraph.

We revised the manuscript and corrected the issue regarding short paragraphs.

Round 2

Reviewer 1 Report

Comments and Suggestions for Authors

All comments have been thoroughly addressed. I extend my gratitude to both the authors and editors for taking my opinions into consideration during the review of this manuscript.

Reviewer 2 Report

Comments and Suggestions for Authors

the authors provided enough explanations regarding my previous comments